# Correlations between the Phylogenetic Relationship of 14 *Tulasnella* Strains and Their Promotion Effect on *Dendrobium crepidatum* Protocorm

Jiayi Zhao, Zhenjian Li, Siyu Wang, Fu Yang, Lubin Li and Lei Liu *

Key Laboratory of Silviculture of the State Forestry Administration, Research Institute of Forestry, Chinese Academy of Forestry, Beijing 100091, China
* Correspondence: liulei519@caf.ac.cn

**Abstract:** The compatibility of mycorrhizal fungi with the early growth stage of orchids is essential for their growth. In this study, the compatibility and promotion effects of 14 *Tulasnella* strains from different hosts were studied by co-culturing them with the protocorms of *Dendrobium crepidatum*, which has high ornamental and economic value in China. The ITS–LSU–SSU–TEF combined sequence analysis divided the 14 strains into three clades belonging to *Tulasnella calospora* (clades A and B) and *Tulasnella asymmetrica* (clade C). All the strains were compatible with *D. crepidatum* protocorms within 90 d of the co-culture. Strain T12 in Clade A had a significantly higher ($p < 0.05$) effect on the biomass and morphology of *D. crepidatum,* and strain T13 in Clade C had a significantly lower ($p < 0.05$) effect than the other strains. Through morphological principal component analysis, we constructed a hierarchical cluster analysis tree, which was consistent with the phylogenetic tree of these 14 strains at the clade level. Orthogonal partial least squares-discriminant analysis showed that these strains have an important effect on the plant height, root number, and length of *D. crepidatum*. The findings of this study will contribute to the identification of *Tulasnella* strains, conservation of *D. crepidatum* resources, and commercial utilization of mycorrhizal technology.

**Keywords:** *Dendrobium crepidatum*; mycorrhizal fungi; *Tulasnella* spp.; molecular identification; symbiosis





## 1. Introduction

Orchid mycorrhizal (ORM) is a mutualistic symbiosis established between orchids and ORM fungi [1]. Orchid seeds are tiny and lack sufficient nutrient reserves, as they are not surrounded by an endosperm [2–4]. Therefore, they obtain carbon, mineral nutrients, and vitamins from their symbiotic partners, which enables them to germinate and develop into a unique seedling structure composed of parenchyma cells called protocorms [5–7]. Subsequently, ORM fungal hyphae enter the protocorm to form coiled complexes called pelotons, and orchids obtain nutrients by digesting the pelotons [5,8,9]. Additionally, the further development of orchids from protocorms to plants also requires the support of ORM fungi [5,10].

Most ORM fungi belong to the 'Rhizoctonia' species complex. Among them, *Tulasnella*, which belongs to the fungal family Tulasnellaceae, is the most predominant ORM fungus found in temperate and tropical regions and has been extensively studied because of its abundant and widespread distribution [11–13]. Therefore, studies on the identification of *Tulasnella* species can better exploit and utilize mycorrhizal resources [14]. With numerous *Tulasnella* species, molecular sequencing provides a powerful method for *Tulasnella* species identification [15,16]. The most important and commonly sequenced fragments are the internal transcribed spacers (ITS) of fungal ribosomal DNA [14,17–20]. In addition, several studies have reported the use of multilocus species delineation to further improve the resolution of *Tulasnella* species identification [21]. Species delimited by *Tulasnella* mainly

include the following loci: nuclear rDNA internal transcribed spacer region (ITS), large sub-unit region (nrLSU) [22–24], small ribosomal subunit (nrSSU) [25], and mitochondrial large rDNA gene (mtLSU) [26,27]. Furthermore, three protein-coding genes are commonly used in the phylogeny of the fungal phylum Basidiomycota: RNA polymerase II largest subunit (RPB1), RNA polymerase II second largest subunit (RPB2), and translation elongation factor 1α (TEF1) [28–30]. Multilocus sequence analysis (MLSA) has broadened previous knowledge of *Tulasnella* species identification.

Notably, not all ORM fungi can promote orchid growth, and fungi that promote orchid growth may not be long-lasting. The compatibility between the orchid and the ORM fungi may change with the growth of orchids [1,3,9,31–33], and only orchid-compatible mycor-rhizal fungi can support the development of orchids into seedlings [3]. This symbiotic pattern is particularly evident in *Dendrobium* plants. *Dendrobium* is one of the largest genera in the orchid family and has been the focus of the cut flower and pharmaceutical indus-tries for decades [34–36]. Several symbiotic studies have been conducted on *Dendrobium* plants and ORM fungi. In previous co-culture studies, *Tulasnella* strains compatible with *Dendrobium* plants rapidly promoted seed germination, protocorm growth, and develop-ment into seedlings. In contrast, *Tulasnella* strains incompatible with *Dendrobium* plants stimulated the seeds to germinate, but the protocorms did not grow and develop into seedlings [32]. For example, the *Tulasnella* sp. SSCDO-5 can promote the development of *D. officinale* from the protocorm to the seedling stage, whereas *Tulasnella* sp. FDd1 can only promote the germination of *D. officinale* seeds and cannot form protocorms [3]. Similar findings have been reported for other *Dendrobium* plants, such as *D. exile* [37], *D. monil-iforme* [38], and *D. chrysotoxum* [39]). The compatibility between *Tulasnella* strains and *Dendrobium* plants changes with the growth of *Dendrobium*, therefore, effective strains of *Dendrobium* at different growth stages can be specifically screened [33,40]. In addition, determining the effects of different *Tulasnella* species on *Dendrobium* plant growth will help meet the needs of artificial propagation of *Dendrobium* and promote the development of the *Dendrobium* industry [37].

*Dendrobium crepidatum*, a traditional Chinese medicine in Yunnan Province, is an epiphytic orchid medicine (TCM) in South Asia [41]. The active components of *D. crepidatum* mainly include indole–azine alkaloids [42,43], flavonoids [44], and bioactive secondary metabolites [45]. Furthermore, *D. crepidatum* has recently attracted attention because of its high medicinal value. However, compared to studies on the active ingredients of *D. crepidatum*, there have been few studies on its symbiotic relationship with ORM fungi [46]. In this study, 14 *Tulasnella* strains were co-cultured with *D. crepidatum* protocorms to establish a symbiotic relationship. We aimed to answer the following questions: (1) Will polygenic MLSA identification of *Tulasnella* strains improve resolution compared to using ITS sequences alone? (2) What is the effect of the 14 *Tulasnella* strains on the growth of *D. crepidatum* protocorms? (3) What is the association between the 14 *Tulasnella* strains' phylogenetic relationship and the promotion efficiency of *D. crepidatum*?

## 2. Materials and Methods

### 2.1. Fungal Isolation and Identification

Purified fungal strains were obtained from previous studies, and the sources of the plant samples are *Cymbidium mannii*, *Epidendrum radicans*, *Cymbidium faberi*, *Cymbidium goeringii*, and *Epidendrum radicans* (Table 1). The fungal strains were cultured on potato dextrose agar (PDA; BD Difco 213400, Pine Hill, NY, USA). The colony morphology was recorded, and the hyphae of the fungi were cultured on PDA for 8 days at 28 °C. The fungal mycelia were frozen using liquid nitrogen, and DNA was extracted using the E.Z.N.A$^{TM}$ Fungal DNA Miniprep Kit (D3390-01, Omega, Guangzhou, China), according to the manufacturer's instructions. This study used the primers ITS1 and ITS4 [47] for the nuclear ITS region and the primers LROR and LR5 [30] for the nuclear LSU-rDNA region. The primers for the SSU and TEF regions are listed in Table S1. The ITS, LSU, SSU, and TEF sequences were compared with those in the NCBI GenBank database using the basic local

alignment search tool (BLAST). Subsequently, the ITS, LSU, SSU, and TEF sequences of the 14 strains were registered in GenBank (Table 1).

**Table 1.** NCBI registration number of fungal gene and orchid source of isolated fungus.

| Isolates | Species | Original Plants | NCBI Accession Number | | | |
|---|---|---|---|---|---|---|
| | | | ITS | LSU | SSU | TEF |
| T1 | *T. calospora* | *C. mannii* | OM672252 | OM728181 | OP537825 | OP819646 |
| T3 | *T. asymmetrica* | *E. radicans* | OM672253 | OM728182 | OP537826 | OP819647 |
| T4 | *T. calospora* | *C. mannii* | OM672254 | OM728183 | OP537827 | OP819648 |
| T5 | *T. calospora* | *C. faberi* | OM672255 | OM728184 | OP537828 | OP819649 |
| T6 | *T. calospora* | *C. faberi* | OM672256 | OM728185 | OP537829 | OP819650 |
| T7 | *T. asymmetrica* | *C. mannii* | OM672257 | OM728186 | OP537830 | OP819651 |
| T9 | *T. calospora* | *C. mannii* | OM672258 | OM728187 | OP537831 | OP819652 |
| T10 | *T. calospora* | *C. goeringii* | OM672259 | OM728188 | OP537832 | OP819653 |
| T11 | *T. calospora* | *C. goeringii* | OM672260 | OM728189 | OP537833 | OP819654 |
| T12 | *T. calospora* | *C. goeringii* | OM672261 | OM728190 | OP537834 | OP819655 |
| T13 | *T. asymmetrica* | *C. goeringii* | OM672262 | OM728191 | OP537835 | OP819656 |
| T14 | *T. calospora* | *E. radicans* | OM672263 | OM728192 | OP537836 | OP819657 |
| T24 | *T. asymmetrica* | *E.radicans* | OM672264 | OM728193 | OP537837 | OP819658 |
| T25 | *T. asymmetrica* | *C. mannii* | OM672265 | OM728194 | OP537838 | OP819659 |

For phylogenetic analysis, 21 strains of *Tulasnella* sp. were constructed using ITS sequences, with *Sebacina vermifera* isolate K251 and *Sebacina vermifera* isolate FFP337 classified as outgroups. Phylogenetic trees were constructed using five genome sequences and ITS–LSU–SSU–TEF combined sequences, with the *Trichoderma reesei* genome as an outgroup.

Finally, 14 polygenic combination sequences were used to construct an evolutionary tree to compare the effects of the relationships between the fungi and the growth of protocorms. BioEdit 7.2 (https://bioedit.software.informer.com/7.2/, accessed on 8 November 2022) was used to calculate the sequence identity and sequence concatenation. All of the sequences were analyzed using maximum likelihood, applying the rapid bootstrapping algorithm for 1000 replications in MEGA7 [48]. Clades with bootstrap values (BS) $\geq 50\%$ were considered significantly supported [49].

### 2.2. Protocorm Formation

*Dendrobium crepidatum* seed capsules were collected from the greenhouse of the Chinese Academy of Forestry (Beijing, China) at day/night temperatures of 24 °C/18 °C, relative air humidity of 65–75%, and 8-h day/16-h night photoperiod. Four *D. crepidatum* mature capsules were collected from the greenhouse prior to dehiscence (Figure 1). After collecting the seed capsules, the first scrub was placed using a soft brush under running tap water to remove dirt and other stains. Each capsule was then cleaned thrice with 75% ethanol, sterilized with 5% NaClO for 5–10 min, and rinsed thrice in sterile distilled water. Finally, the capsules were cut lengthwise, and dusty ripe seeds were picked [50]. After sterilization, as described above, approximately 100–150 seeds were added to a culture flask (glass bottles) containing Murashige and Skoog (MS) medium [51]. The cultures were placed in a growth room under a 12/12-h photoperiod at $25 \pm 1$ °C with a relative air humidity of 65–75%.

### 2.3. Symbiotic Culture of Protocorms

Protocorms are used for co-culture when they develop to stage 2 [39] (rupture of testa, appearance, and extension of the protomeristem). Protocorms with consistent growth were inoculated in 2.0 g/L OA medium [50], and an approximately 1 cm$^3$ cube of each fungal culture was inverted onto the surface of the OA medium using an inoculating needle. The control treatment group contained no fungal inoculum. The Petri dishes were sealed with Parafilm (Bemis, Neenah, WI, USA) and incubated at 24 °C/18 °C day/night temperatures and 65–75% relative humidity. Each treatment comprised 10 replicates. Symbiotic cultures were placed in a growth room under a 12/12-h photoperiod at $25 \pm 1$ °C, with a relative air humidity of 65–75%, as described previously. The plates were sealed with Parafilm (Bemis,

Neenah, WI, USA) and maintained at day/night temperatures of 24 °C/18 °C and relative air humidity of 65–75%.

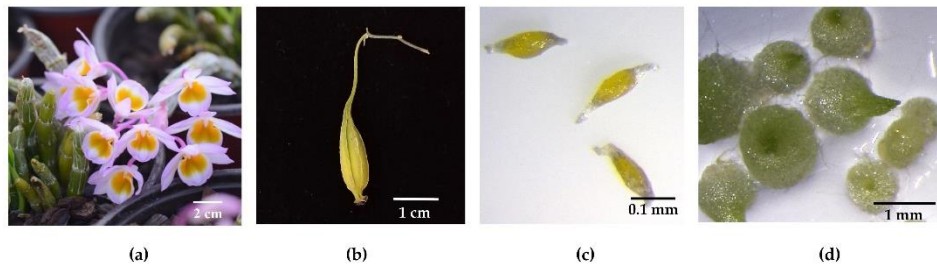

**Figure 1.** Plant material of *D. crepidatum*. (**a**) The flowers of *D. crepidatum*; (**b**) The capsules of *D. crepidatum*; (**c**) Seeds swollen after absorbing water; (**d**) The protocorms of *D. crepidatum*. Bar: (**a**) 2 cm, (**b**) 1 cm, (**c**) 0.1 mm, (**d**) 1 mm.

*2.4. Assessment of the Fungal Capacity to Promote Protocorm and Seedlings Growth*

Protocorm growth was observed daily under a stereomicroscope (Nikon HFX, Tokyo, Japan) and light microscope (OLYMPUS BX51, Tokyo, Japan). Trypan blue staining was performed to examine fungal colonization [52]. A concentration of 0.4% trypan blue stain was used for the study. Data were subsequently collected on days 30, 60, and 90 after sowing to determine the average fresh and dry weight growth rates. When measuring fresh weight, 30 symbiotic materials were randomly selected for each sampling, and the average value was calculated three times. To measure the dry weight, 30 symbiotic materials were randomly selected and dried at 60 °C until the weight remained constant. The samples were recorded, which was repeated three times. The average growth rate was determined as follows: (weight after inoculation − weight before inoculation amount)/weight before inoculation × 100% [53].

At 90 days of co-cultivation, the plant height, stem diameter, root length, leaf length, and leaf width were measured using a Vernier caliper, and the root number and leaf number were recorded [54]. Ten symbionts were randomly selected during the measurement, and the average value was obtained and the measurement was repeated three times. The relative leaf area is a product of the leaf length and width [55]. SIMCA 17 software (https://www.sartorius.com/en/products/process-analytical-technology/data-analytics-software/mvda-software/simca, accessed on 8 November 2022) was used to perform principal component analysis (PCA), hierarchical clustering analysis (HCA), and orthogonal partial least squares-discriminant analysis (OPLS-DA) of the morphological parameters. In addition, HCA was performed on the PCA results, and the HCA tree of principal component 1 (PC1) and principal component 2 (PC2) was constructed via sorting by index using Ward calculation.

*2.5. Statistical Analysis*

All the experiments were performed in a completely randomized design. One-way ANOVA was used, and the significant differences between the treatments were assessed using least significant difference (LSD) multiple comparison tests ($p < 0.05$). All the statistical analyses were performed using IBM SPSS Statistics 25.0 (IBM Corporation, Armonk, NY, USA). GraphPad Prism 9 (www.graphpad.com, accessed on 26 September 2022)and R Studio software (https://posit.co/downloads/, accessed on 9 November 2022).

**3. Results**

*3.1. Tulasnella Identification and Phylogenetic Analyses*

The BLAST results of the ITS sequences in the GenBank database showed that all 14 strains were *Tulasnella* (Table S2). The phylogenetic trees constructed based on ITS sequences showed that the nine strains were clustered with *T. calospora* and divided into two branches. The first branch (Clade A) consisted of five strains (T5, T12, T10, T4, and T1),

and the second branch (Clade B) consisted of four strains (T14, T6, T9, and T11) (Figure 2). Meanwhile, five strains (T24, T25, T3, T7, and T13) formed a separate phylogenetic lineage within *T. asymmetrica*, named Clade C (Figure 2).

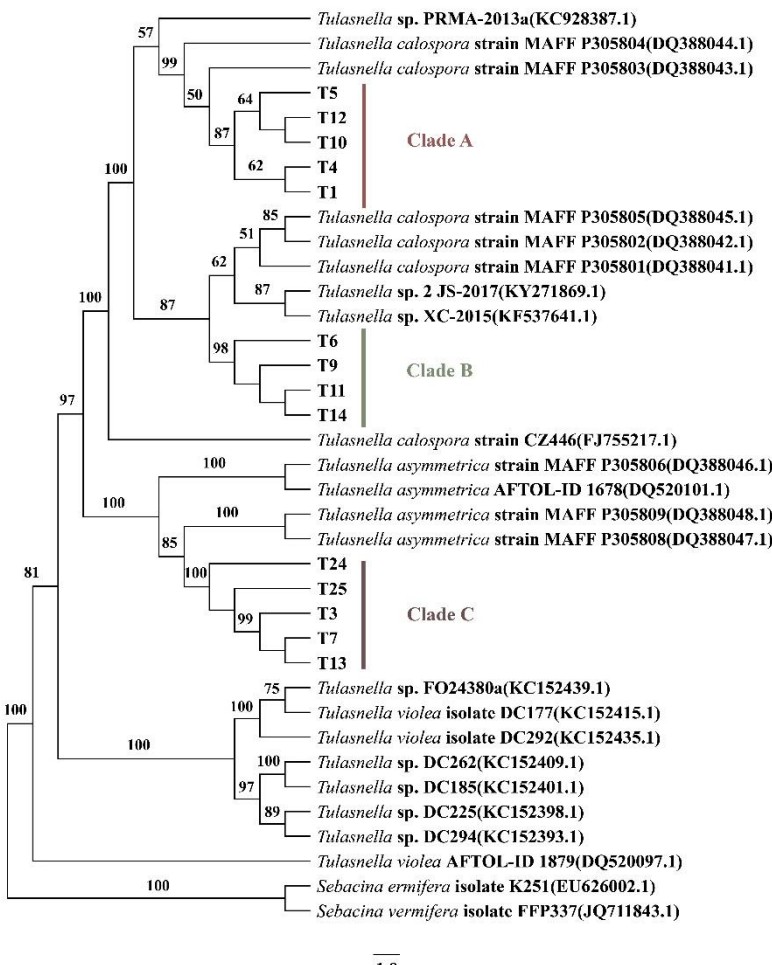

**Figure 2.** Maximum likelihood trees of ITS sequences of *Tulasnella* sp. The sequence of *Sebacina vermifera* isolate FFP337 and *Sebacina vermifera* isolate K251 was used as an outgroup. Only bootstrap values (based on 1000 replications) ≥ 50 are shown. Scale, 1.0 nt substitutions per site.

The comparison results of the ITS sequence consistency showed that the sequence identity of the 14 strains ranged from 56.5–100% (Figure S1). Notably, the sequence identity of strains T3 and T7, and T6 and T9 was 100% (Figure S1). Further identification using ITS–LSU–SSU–TEF combined sequences (Figure S2) showed that the sequence identities of strains T3 and T7, and T6 and T9 changed to 99.7% and 99.9%, respectively (Figure S1). These results suggest that polygenic analysis alters the resolution compared to ITS sequences alone. Furthermore, the colony and mycelium morphology showed no significant difference in the colony and hyphae of the three strains (strain T12 in Clade A, T9 in clade B, and T13 in Clade C). The mycelia of these three strains near the inoculated agar blocks had distinct concentric circles and rectangular branches in the mycelia (Figure 3).

### 3.2. Compatibility between Tulasnella Stains and D. crepidatum Protocorms

After co-culturing the fungi with *D. crepidatum* protocorms, all *D. crepidatum–Tulasnella* symbiotic protocorms grew and developed into seedlings. At 30 days of co-culture, the protocorm in symbiosis with *Tulasnella* began differentiating into its first leaf (Figure 4a). After 60 days of co-culture, the symbiotic protocorms developed into seedlings with two leaves and one root (Figure 4b). Finally, after 90 days of co-culture, the symbiotic pro-

tocorms further developed into seedlings with an average of four leaves and two roots (Figure 4c). These results indicate that the 14 *Tulasnella* strains used in this study and *D. crepidatum* protocorms are compatible (Figure 4a–c). Meanwhile, successful colonization of the symbiotic protocorms by the 14 *Tulasnella* strains was observed under stereomicroscopy (Figure 4d–f) and light microscopy (Figure 4g–i). Furthermore, the results of trypan blue staining showed that the strains colonized the parenchyma cells of *D. crepidatum* protocorms, and the pelotons were apparently visible when co-cultured for 30, 60, and 90 days (Figure 4d–i). Furthermore, all 14 strains belonging to *T. calospora* (in Clade A and Clade B) and *T. asymmetrica* (in Clade C) were compatible with *D. crepidatum* protocorms.

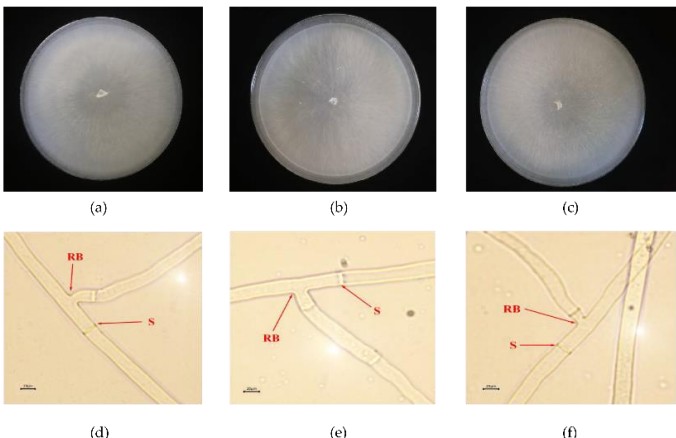

**Figure 3.** The mycelium morphological characteristics of three clades strains. (**a**) Community morphology of T12 (Clade A); (**b**) Community morphology of T9 (Clade B); (**c**) Community morphology of T13 (Clade C); (**d**) Mycelium morphology of T12 (Clade A); (**e**) Mycelium morphology of T9 (Clade B); (**f**) Mycelium morphology of T13 (Clade C). RB: Rectangular branch; S: Septum. The Petri dishes are 9 cm in diameter.

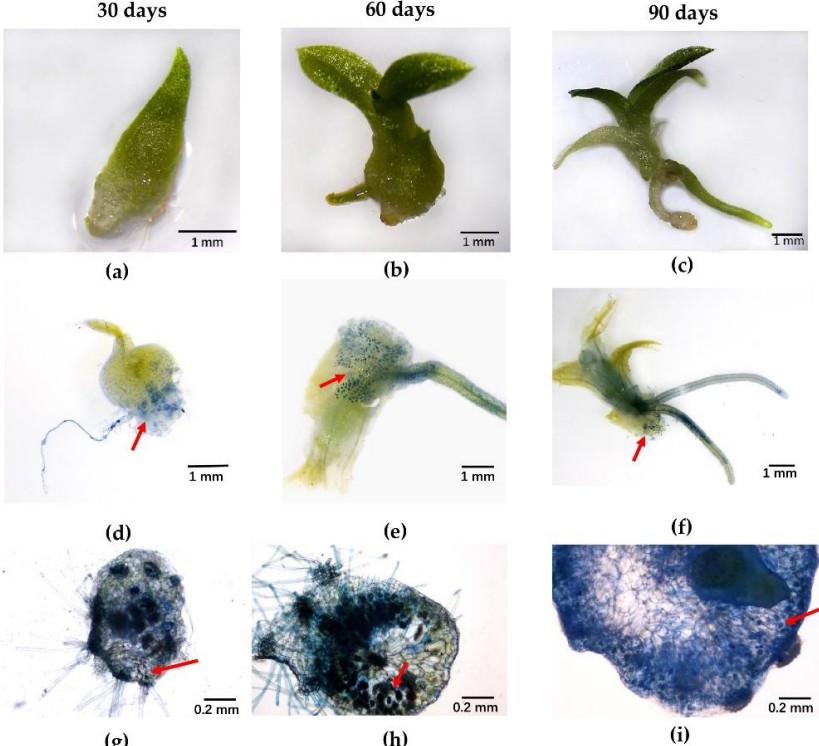

**Figure 4.** The colonization of strain T12 in *D. crepidatum* protocorms during 90 days of co-culture. The

growth of the *D. crepidatum* protocorms into seedlings in co-culture at 30 days (**a**), 60 days (**b**), and 90 days (**c**). Pelotons were observed in a stereomicroscope using trypan blue staining at 30 days (**d**), 60 days (**e**), and 90 days (**f**) of co-culture; Cross-section of the symbionts that were observed in the light microscope using trypan blue staining at 30 days (**g**), 60 days (**h**), and 90 days (**i**) of co-culture. Bar: (**a**–**f**) 1 mm, (**g**–**i**) 0.2 mm. The red arrows point to the pelotons.

### 3.3. Effects of different Tulasnella Strains on the Biomass of the Symbionts

The biomass parameters of the fungus–protocorm symbiont, including the average fresh weight growth rate and the average dry weight growth rate, represent the growth status of fungus–protocorm symbiont. The average fresh weight growth rate and average dry weight growth rate of the 14 symbionts were significantly higher ($p < 0.05$) than those of the control group on days 30, 60, and 90 after co-culture, and this growth-promoting effect lasted for 90 days (Figure 5, Figure S3, Tables S3 and S4). This result indicates that the 14 *Tulasnella* strains are consistently and stably compatible with *D. crepidatum* protocorms and promote their growth.

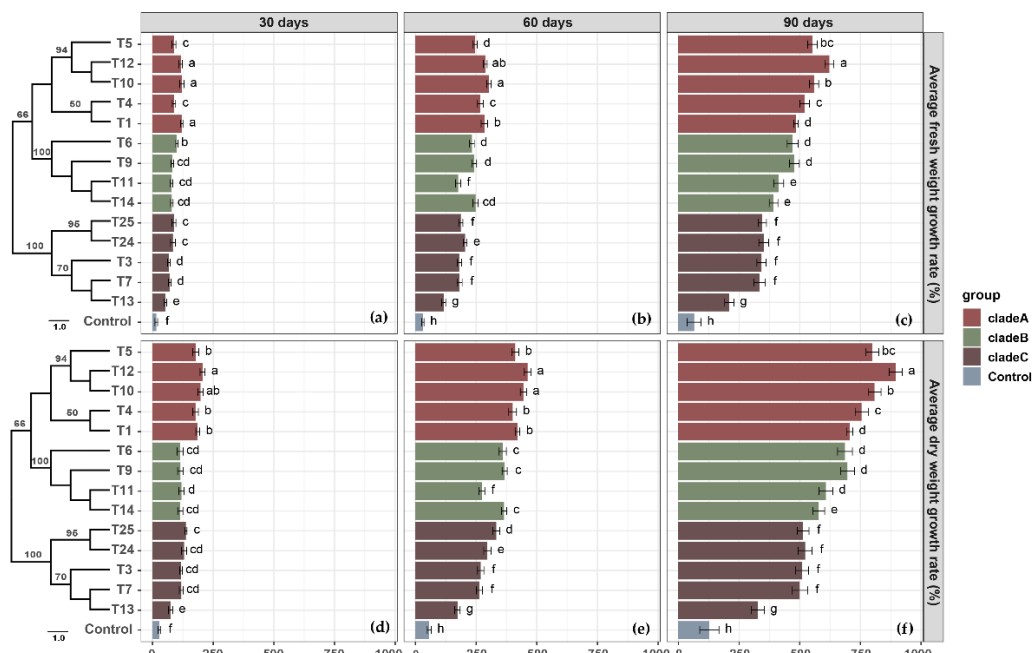

**Figure 5.** Comparison of the average fresh and dry weight growth rates of 14 fungi on the 30th, 60th, and 90th days of symbiosis. (**a**) The average fresh weight growth rate on day 30 after co-culture. (**b**) The average fresh weight growth rate on day 60 after co-culture. (**c**) The average fresh weight growth rate on day 90 after co-culture. (**d**) The average dry weight growth rate on day 30 after co-culture. (**e**) The average dry weight growth rate on day 60 after co-culture. (**f**) The average dry weight growth rate on day 90 after co-culture. Strains with different lowercase letters are significantly different ($p < 0.05$) based on the LSD test.

Meanwhile, the 14 compatible *Tulasnella* strains differed in their promotional effect on *D. crepidatum* protocorms (Figure 5). In detail, during the 90 days of co-cultivation, strain T12 in Clade A had the strongest effect ($p < 0.05$) on promoting the fresh weight (the average growth rates were 116.67 ± 7.02%, 288.02 ± 7.55%, and 622.16 ± 19.06% on days 30, 60, and 90 of co-culture, respectively) and dry weight (206.00 ± 10.15%, 462.67 ± 13.32%, and 895.37 ± 26.27% on days 30, 60, and 90, respectively) of *D. crepidatum* protocorm, whereas strain T13 in Clade C had the weakest effect ($p < 0.05$) on promoting the fresh weight (only 54.01 ± 4.58%, 116.33 ± 8.08%, 210.28 ± 18.78% on days 30, 60, and 90 after co-culture, respectively) and dry weight (74.33 ± 8.62%, 173.02 ± 11.53%, and 327.67 ± 25.88%, on days 30, 60, and 90 after co-culture, respectively) among all 14 strains (Figure 5).

Overall, when cultured for 30 days, three strains of Clade A (T12, T10, and T1) had a significantly higher effect on the fresh weight ($116.67 \pm 7.02\%$, $120.67 \pm 8.51\%$, and $120.67 \pm 5.03\%$ in T12, T10, and T1, respectively) and dry weight ($206.00 \pm 10.15\%$, $198.00 \pm 11.53\%$, and $186.67 \pm 8.08\%$ in T12, T10, and T1, respectively) of the protocorms than the other strains. Furthermore, on days 60 and 90 of the co-culture, the growth-promoting effect of the *Tulasnella* strain in clade A was evidently higher than that of other strains in Clade B and Clade C, whereas the five strains in Clade C were significantly lower than those of the other strains. Overall, these results indicate that the growth-promoting effects of the strains may be related to their evolutionary relationship.

### 3.4. Effects of Different Tulasnella Strains on the Morphology of the Symbionts

Trypan blue staining results showed that pelotons were detected in all 14 symbionts after 90 days of co-culture, and the symbionts grew from protocorms to seedlings (Figure 4f,i). Therefore, to further understand the role of various strains in protocorm growth, we investigated phenotypic data from six 90-day symbionts. On day 90 of symbiosis, the morphology of the symbionts was significantly better than the non-symbiotic control group (Figure 6). Further morphological measurements showed that, compared with that of the control group, the plant height, stem diameter, root length, root number, and leaf number of all 14 symbionts were significantly increased ($p < 0.05$) (Figure 7a–e, Table S5). However, the relative leaf areas of the four symbionts (strains T24, T3, T7, and T13) were not significantly different from that of the control (Figure 7f, Table S5). Similar to the previous results on biomass, among different *Tulasnella* strains, strain T12 was significantly better ($p < 0.05$) than the other strains in promoting the plant height ($8.68 \pm 0.16$ mm), stem diameter ($2.39 \pm 0.05$ mm), root length ($15.14 \pm 0.46$ mm), root number ($2.22 \pm 0.19$), leaf number ($5.00 \pm 0.33$), and relative leaf area ($7.26 \pm 0.72$ mm$^2$) of symbionts (Figure 7). By contrast, strain T13 had the weakest ($p < 0.05$) promotional effects on plant height ($5.55 \pm 0.37$ mm), stem diameter ($1.89 \pm 0.04$ mm), root length ($2.97 \pm 0.28$ mm), root number ($0.89 \pm 0.19$), and leaf number ($2.45 \pm 0.16$) (Figure 7). Compared with that of other strains, the promotional effect of strain T5 on symbiont plant height ($8.50 \pm 0.27$ mm) and stem thickness ($8.50 \pm 0.27$ mm) (Figure 7a,c) was not significantly different ($p > 0.05$) from that of the dominant strain T12, whereas its promotional effect on the root length ($8.97 \pm 0.67$ mm) and root number ($1.56 \pm 0.19$ mm) was significantly different ($p < 0.05$) from that of the dominant strain T12 (Figure 7b,e).

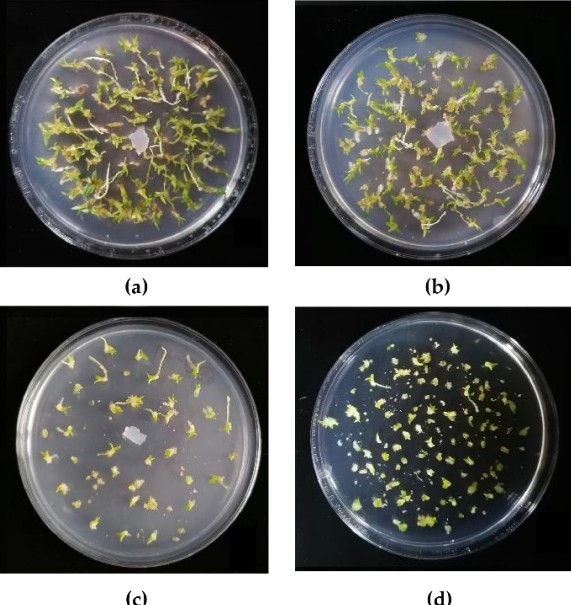

**Figure 6.** Morphology of the symbionts on day 90. (**a**) Strain T12 in Clade A; (**b**) Strain T9 in Clade B; (**c**) Strain T13 in Clade C; (**d**). Control. The Petri dishes are 9 cm in diameter.

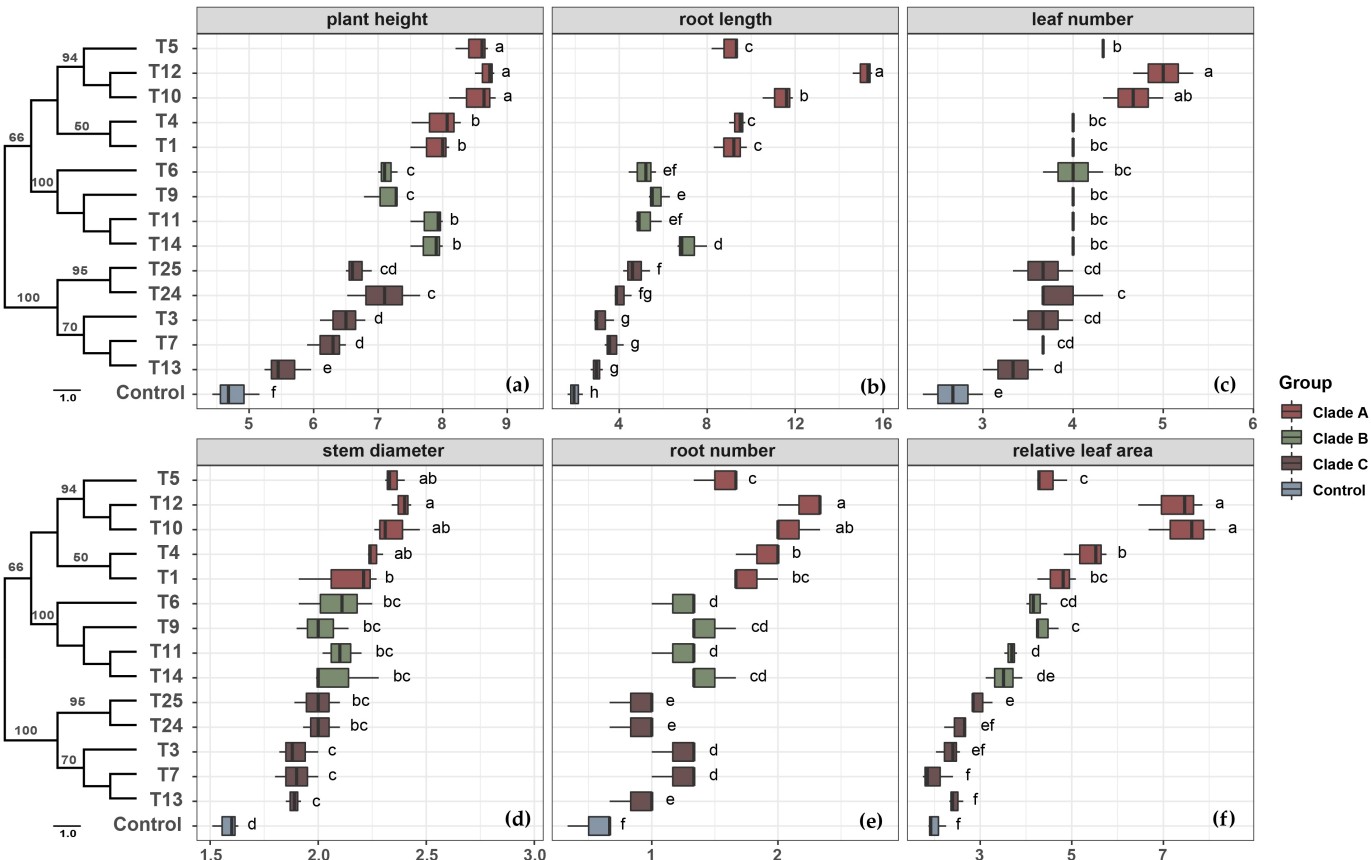

**Figure 7.** The growth of different fungi and *D. crepidatum* protocorm on day 90 of symbiosis. (**a**) The average plant height of symbionts; (**b**) The average root length of symbionts; (**c**) The average leaf number of symbionts; (**d**) The average stem diameter of symbionts; (**e**) The average root number of symbionts; (**f**) The relative leaf area of symbionts. Strains with different lowercase letters are significantly different ($p < 0.05$) based on the LSD test.

Notably, strains T12 and T10 (in Clade A) promoted the six morphological parameters to a larger extent than the other strains. In contrast, strains T3, T7, and T13 (in Clade C) had a weaker promotion effect. These results further suggest that the promoting efficiency may be related to the genetic relationship among *Tulasnella* strains.

### 3.5. Association between Relationship and Symbiotic Effects among Different Tulasnella Strains

Since previous results have found that the promoting efficiency of symbiont growth by different *Tulasnella* strains may be related to the relationship between the strains, PCA and OPLS-DA methods were used to further analyze the association between them. The PCA for the six morphological parameters showed that the contribution rate of PC1 and PC2 was 90.6% and 4.49%, respectively, and the sum of the two principal components was greater than 95% (Figure 8d). Simultaneously, the 14 *Tulasnella* strain symbionts were divided into three clades in PC1. The fungi within the three clades in the PCA results are consistent with those within the three clades in the phylogenetic tree constructed using ITS (Figure 2) and polygenes (Figure S2). Four symbionts (strains T12, T10, T4, and T1) in Clade A were distinguished in PC1, strain T5 in Clade A was distinguished in PC2, and strain T12 had the highest contribution rate in PC1 (Figure 8d). The four symbionts (strains T6, T9, T11, and T14) in Clade B were distinguished in PC2. The other symbionts (strains T24, T25, T3, T7, and T13) in Clade C were distinguished by PC1 and PC2 (Figure 8d). Furthermore, HCA was performed on the PCA results, and the HCA tree divided the strains into three clades. Compared with the phylogenetic tree, the HCA tree was consistent with the phylogenetic tree at the clade level (Figure 8a,b). Further analysis of PC1 in the PCA revealed that the

contribution rate of different fungi showed a downward trend from clade A to clade C, and strain T12 also had the highest contribution rate (Figure 8c).

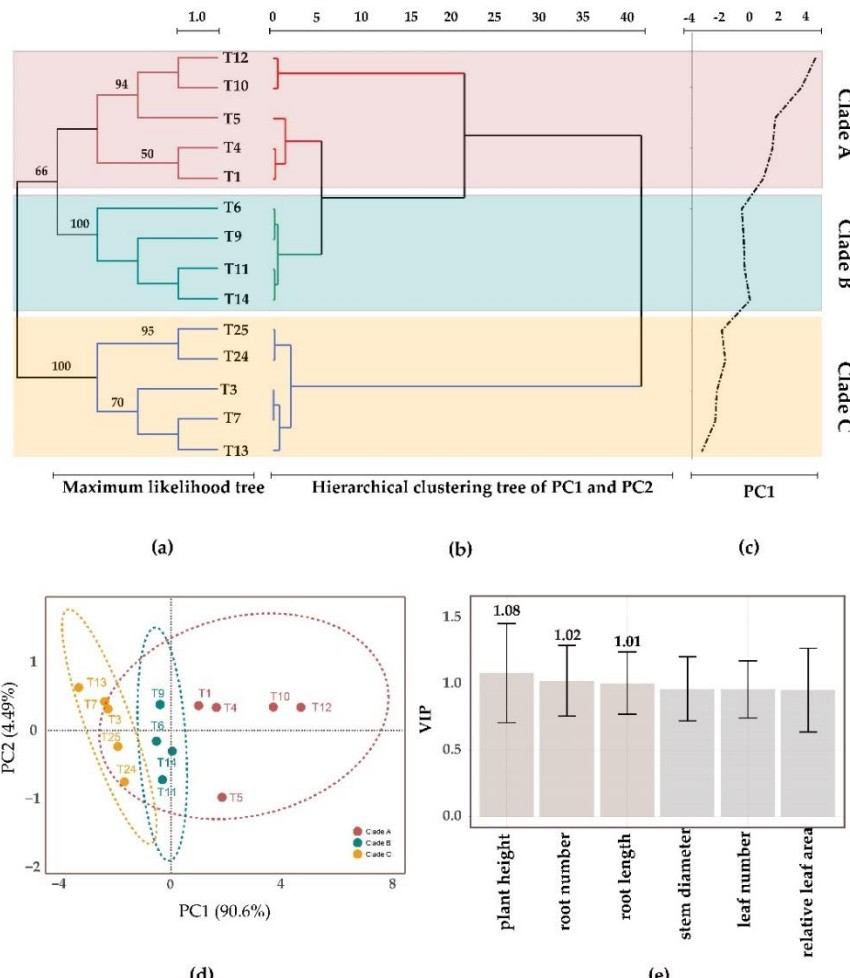

**Figure 8.** PCA and OPLS-DA of different fungal symbionts. (**a**) Maximum likelihood trees of ITS sequences of *Tulasnella* sp.; (**b**) Hierarchical clustering analysis of six morphological parameters; (**c**) PC1 analysis in a line plot; (**d**) Principal component analysis of two principal components (PC1 and PC2); (**e**) VIP values of six parameters in orthogonal partial least squares-discriminant analysis.

In addition, the OPLS-DA model was used to analyze the contribution of the 14 strains to six morphological parameters. To verify the reliability of the OPLS-DA model, 200 permutation tests were performed on two significant components. The R2Y intercept was 0.117 (<0.2) and the Q2 intercept was −0.566 (<0.05), indicating that the OPLS-DA model used was reliable (Figure S4). Furthermore, using the OPLS-DA model, the VIP values of the plant height (1.08), root number (1.02), and root length (1.01) were greater than 1.0, indicating that these three parameters were the key variables (Figure 8e).

## 4. Discussion

Orchids are heterotrophic in the early stages of development and require ORM fungi to provide nutrients that are likely extracted from soil organic matter for their growth and development [7,56,57]. *Dendrobium crepidatum* is a perennial epiphyte distributed in southern China, where it is known as "Shi-Hu". *Dendrobium crepidatum* can moisten the lungs, relieve cough, clear deficiency heat, and nourish the stomach [41]. *Tulasnella* is extremely important for orchid mycorrhizas as ORM fungi. Therefore, it is vital to study the symbiosis between the *Tulasnella* species and *D. crepidatum* to conserve *D. crepidatum* resources.

Previous studies have shown that *Tulasnella* species fruiting bodies are usually absent and difficult to induce in culture, which are less helpful for *Tulasnella* species identification [58–62]. Therefore, recent studies have focused on using molecular sequencing to identify *Tulasnella* species [15,16,23,63–68]. Polygenic analysis has a higher resolution than ITS single gene analysis, and fungal species taxonomy should be based on the phylogeny of core genes with strong phylogenetic signals, such as ITS regions and at least one protein gene, including TEF or RPB2 [21,22,24,28,29,68]. Overall, multilocus DNA sequence datasets typically contain two to six genes depending on the research needs [16,21,69]. However, recent polygenic analyses of *Tulasnella* have focused on the identification of new species of *Tulasnella* fungi and have been used less for subspecies-level identification [15,16,24]. Our study successfully divided the 14 strains into two species, *T. calospora* and *T. asymmetrica*, using ITS sequences. We used four genes for MLSA identification for each intraspecific strain and further altered the sequence consistency of T3, T7, T6, and T9 (Table S3). These results are consistent with those of previous changes in resolution through MLSA [29,30] and provide a reference for the identification of *Tulasnella* subspecies levels. The next step should be to increase the number of genes to further improve the resolution and, thus, better exploit mycorrhizal resources.

Not all mycorrhizal fungi are compatible with *Dendrobium,* and the compatibility of the fungi may change with the growth of mycorrhizal symbionts [3,39,70]. Moreover, persistent observation of pelotons is a sign of fungi–*Dendrobium* compatibility [3]. For example, a study in which six *Tulasnella* species were co-cultured with *D. moniliforme* seeds showed that only two *Tulasnella* species symbionts were still alive after 120 days of co-culture because other *Tulasnella* species symbionts could not develop into seedlings from protocorms [38]. In other species of *Dendrobium*, mycorrhizal fungi (e.g., *Tulasnella* species) can live symbiotically with *Dendrobium* seeds until the formation of protocorms; however, they often change, and either the compatible fungi continue to promote protocorm differentiation to form seedlings or the protocorms stop growing or even die [3,4,37–39]. The same phenomenon occurs in the symbiosis between fungi and other orchids, such as *Arundina graminifolia* [52] and *Serapias vomeracea* [17]. In our study, the 14 *Tulasnella* strains, belonging to *T. calospora* and *T. asymmetrica,* are compatible with *D. crepidatum* protocorms. This is consistent with previous studies, thereby showing that compatible fungi can successfully differentiate protocorms into seedlings and can continuously observe pelotons (Figures 5 and 6).

In response to these phenomena, it has been hypothesized that during the transition from protocorms to seedlings, protocorms may require a higher level of carbon sources in terms of quantity and quality. Therefore, a better functional match with the fungus may result in enhanced carbon flow, thereby producing more robust seedlings [38]. Furthermore, under natural conditions, changes in mycorrhizal fungal compatibility can allow orchids to adapt to various physiological changes during seedling growth, including partial or complete autotrophy, increased transpiration, or environmental fluctuations [71]. In addition, compatibility between fungi and their hosts probably depends on the developmental stage, which may contribute to narrowing the host distribution range [70]. Therefore, the dynamic change in compatibility between *Tulasnella* species and *Dendrobium* can be regarded as an advanced survival strategy, and it is more intuitive and effective to study this compatibility during the protocorm period.

Among compatible *Tulasnella* strains, different strains have varied effects on seed germination, protocorm formation, and seedling development of *Dendrobium* species [31,32,37,40,72]. Moreover, the evaluation indicators include biomass (fresh and dry weight) and various parameters related to the morphology (plant height, root length, and leaf number) of symbionts [33,34,40,73–75]. For example, *T. calospora* strains S6 and S7 can support the development of *D. officinale* protocorms into seedlings, unlike S4. Meanwhile, for the protocorm formation rate with the second leaf, the symbionts inoculated with *T. calospora* strain S7 (59.17 ± 8.78%) were higher than those inoculated with S6 (1.35 ± 1.09%). Furthermore, S7 was better than S6 in promoting the fresh weight and root number of symbionts. In

contrast, S6 was better than S7 regarding the number of tillers of the symbionts and the total crude polysaccharide content in the stem [33]. The same phenomenon has been observed in other studies; for example, among the three strains of *T. calospora*, JM, TG1, and TG3, strain JM had the highest promotion efficiency ($p < 0.05$) on *D. officinale* protocorm formation and seedling development, followed by TG3, and TG1 was not significantly different ($p < 0.05$) from the control group [32]. Furthermore, the phylogenetic tree of the three strains based on ITS-5.8S sequences revealed that JM was on one branch, whereas TG1 and TG3 were in two clades of another branch [32]. These studies suggest that the effectiveness of fungi in promoting *Dendrobium* growth is related to the species and intraspecific phylogenetic relationships of the fungi.

Some studies have explained the reason why different *Tulasnella* strains have varied effects on orchid germination and development may be because *Tulasnella* species are different at the species level and this phenomenon must be studied via precise molecular identification [17]. Meanwhile, the inherent differences between fungi in terms of symbiotic compatibility have a greater impact on mycorrhizal symbiosis than previously speculated. The differences in symbiotic compatibility between *Tulasnella* strains did not reflect their phylogenetic relationships based on fungal ITS sequences because *Tulasnella* ribosomal DNA sequences are extremely variable, and the phylogenetic relationships they present may be inaccurate [75].

In our study, among the 14 *Tulasnella* strains, *T. calospora* strain T12 was the most effective symbiont for promoting the growth of *D. crepidatum* protocorm, whereas *T. asymmetrica* strain T13 had the lowest efficacy (Figures 5–8). The PCA of six morphological parameters revealed that the contribution rates of the 14 strains were different and could be well distinguished, and the names of the strains in the three clades of the hierarchical cluster were consistent with those of the phylogenetic tree (Figure 8). The growth promotion efficiency of the 14 strains decreased from strain T12 in Clade A to strain T13 in Clade C, which proves that for *Tulasnella* and *D. crepidatum*, the phylogenetic relationship between fungi was closely related to the promotion efficiency of symbiotes; and the growth promotion efficiency of *T. calospora* strains was better than that of *T. asymmetrica* for *D. crepidatum*. In addition, although the host plants are different, the two pairs of strains, T3 and T7 and T6 and T9, have a close genetic relationship in terms of identifying the ITS sequence and ITS–LSU–SSU–TEF polygenic sequence and also have similar symbiotic effects for *D. crepidatum*. Although the mechanism of compatibility and the varied symbiotic effectiveness between *Dendrobium* plants and *Tulasnella* strains are still unclear, they may be related to hormones and lignin degradation activity [1,70,76]. Therefore, transcriptomic and secretory tools may be used to further investigate compatibility patterns and symbiotic mechanisms.

## 5. Conclusions

This study identified 14 strains from different hosts as *T. calospora* and *T. asymmetrica*, where *T. calospora* contains clade A and clade B, and *T. asymmetrica* contains clade C. ITS–LSU–SSU–TEF polygene analysis can further improve identification resolution compared to using ITS sequences alone. All 14 strains of *T. calospora* and *T. asymmetrica* were compatible with the *D. crepidatum* protocorm. The protocorms inoculated with *T. calospora* strain T12 in clade A showed optimal fresh and dry matter biomass and morphological parameters. The *T. asymmetrica* strain T13 in clade C had a significantly lower ($p < 0.05$) promotional effect than the other strains. These fungal strains tested in co-culture have different effects on the growth and development of protocorms, and the phylogenetic tree of 14 strains is consistent with the morphological hierarchical clustering tree at the branching level, suggesting that the phylogenetic relationship of 14 *Tulasnella* strains correlates with their promotion effect on the *D. crepidatum* protocorm. These results may pave the way for further research on the relationship between mycorrhizal fungi and *Dendrobium* plants and help to protect *D. crepidatum* resources.

**Supplementary Materials:** The following supporting information can be downloaded at: https://www.mdpi.com/article/10.3390/horticulturae8121213/s1, Figure S1: Sequence consistency across different strains; Figure S2: Maximum likelihood trees of ITS–LSU–SSU–TEF concatenated sequences of *Tulasnella* sp.; Figure S3: Average fresh weight growth rate and dry weight growth rate of symbionts on days 30, 60, and 90 of symbiosis; Figure S4: 200 permutation tests of the OPLS-DA model. Table S1: Primer sequence; Table S2: Blast results of mycorrhizal fungi ITS sequences in the GenBank database; Table S3: The average fresh weight growth rate of protocorms on days 30, 60, and 90 of symbiosis; Table S4: Average dry weight growth rate of protocorms on days 30, 60, and 90 of symbiosis; Table S5: Average growth parameters of protocorms on day 90 of symbiosis.

**Author Contributions:** Conceptualization, L.L. (Lei Liu) and L.L. (Lubin Li); methodology, L.L. (Lei Liu) and J.Z.; overall conception, L.L. (Lubin Li); validation, S.W. and F.Y.; formal analysis, S.W. and F.Y.; investigation, J.Z.; sample resources, Z.L.; data curation, J.Z.; writing—original draft preparation, J.Z.; writing—review and editing, L.L. (Lei Liu); supervision, L.L. (Lei Liu); funding acquisition, L.L. (Lei Liu). All authors have read and agreed to the published version of the manuscript.

**Funding:** This work was funded by the National Natural Science Foundation of China, grant number No. 31800522 and the Fundamental Research Funds for the Central Non-profit Research Institution of Chinese Academy of Forestry CAFYBB2020SZ006.

**Institutional Review Board Statement:** Not applicable.

**Informed Consent Statement:** Not applicable.

**Acknowledgments:** The authors are grateful to Lu Xu (Hunan Mid-Subtropical Quality Plant Breeding and Utilization Engineering Technology Research Center, College of Horticulture, Hunan Agricultural University) for her help with the strain materials and suggestions on writing paper and Yanxia Cheng (State Key Laboratory of Tree Genetics and Breeding, Chinese Academy of Forestry) for her suggestions on experimental methods and data analysis.

**Conflicts of Interest:** The authors declare no conflict of interest.

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
