# Peer review of "Correlations between the Phylogenetic Relationship of 14 Tulasnella Strains and Their Promotion Effect on Dendrobium crepidatum Protocorm"

_horticulturae, doi:10.3390/horticulturae8121213_

Round 1
Reviewer 1 Report
Dear,
The paper “Correlations between the phylogenetic relationship of 14 Tulasnella strains and their promotion effect on Dendrobium crepidatum protocorm” although not very innovative, brings relevant data on symbiotic efficiency for this type of mycorrhiza. However, some adjustments are needed:
- Inform the classifier of all fungi and plants mentioned; this must also be done in the tables;
- Do not start a sentence with the abbreviated genus of organisms;
- Place the bar in Figure 1a;
- In item 2.4, inform the concentration of Trypan blue used;
- In line 274, put p>0.05;
- In line 333, do not use the term “anamorphic”, as it is no longer used.
After the aforementioned adjustments, I consider that the article can be accepted for publication.
Sincerely,
Reviewer 2 Report
The MS's choice of topic is relevant, as the investigated plant has "high ornamental and economic value in China".
The identification of symbiont fungi and the utilization of mycorrhization technology can also bring further success and results, among others in the conservation of D. crepidatum resources.
The research work achieves the goals set in the introduction, it also successfully proves that growth-promoting effects of the strains are related to their evolutionary relationship
Overall, MS presents a very rich and up-to-date literary background, which analyzed with commendable thoroughness in the discussion and compares it with own results.
It is typical of MS and somewhat unusual, but at the same time commendable, that it mostly assigns several references to a single idea or statement.
ad 102-103: Good selection of the outgroup is very important for the success of phylogenetic analyses. Agaricomycetes sp. for markings, a more precise marking is required.
gives Fig. 5. The wording requires clarification. Describe more precisely what growth rate means!
Minor formal and grammatical errors:
Separation: ad 105-106
Highlighting foreign language text: ad 499, 539, 541, 555, 589, 626,
Highlighting Latin genus and species names: ad 517, 571, 573, 622
ad 593.: Correct citation?
Uncompleted citation.: ad 598, 607
